# Physical activity mitigates the link between adverse childhood experiences and depression among U.S. adults

**Michael F. Royer**◯\*, **Christopher Wharton**

Arizona State University, College of Health Solutions, Phoenix, AZ, United States of America

\* mfroyer@asu.edu

## Abstract

### Background

Adverse Childhood Experiences (ACEs) include potentially traumatic exposures to neglect, abuse, and household problems involving substance abuse, mental illness, divorce, incarceration, and death. Past study findings suggest ACEs contribute to depression, while physical activity alleviates depression. Little is known about the link between ACEs and physical activity as it relates to depression among U.S. adults. This research had a primary objective of determining the role of physical activity within the link between ACEs and depression. The significance of this study involves examining physical activity as a form of behavioral medicine.

### Methods

Data from the 2020 Behavioral Risk Factor Surveillance System were fit to Pearson chi-square and multivariable logistic regression models to examine the links between ACEs and depression, ACEs and physical activity, and physical activity and depression among U.S. adults ages 18-and-older (n = 117,204) from 21 states and the District of Columbia, while also determining whether physical activity attenuates the association between ACEs and depression.

### Results

Findings from chi-square analyses indicated that ACEs are related to physical activity ($\chi^2$ = 19.4, df = 1; $p$<0.01) and depression ($\chi^2$ = 6,841.6, df = 1; $p$<0.0001). Regression findings suggest ACEs were linked to depression (AOR = 1.050; 95% CI = 1.049, 1.051). ACEs and physical activity (AOR = 0.994; 95% CI = 0.992, 0.995) and physical activity and depression (AOR = 0.927; 95% CI = 0.922, 0.932) were both inversely related. Physical activity mitigated the link between ACEs and depression (AOR = 0.995; 95% CI = 0.993, 0.996).

### Conclusions

This research addressed a critical knowledge gap concerning how ACEs and physical activity contribute to depression outcomes among U.S. adults. Findings suggest physical activity

**Data Availability Statement:** The data underlying the results presented in the study are available from the CDC (https://www.cdc.gov/brfss/annual_

data/annual_2020.html, https://www.cdc.gov/brfss/annual_data/2020/files/LLCP2020XPT.zip).

**Funding:** The author(s) received no specific funding for this work.

**Competing interests:** The authors have declared that no competing interests exist.

mitigates the effect of ACEs on depression. Future studies should apply physical activity interventions to alleviate depression among U.S. adults with high ACEs.

## Background

Adverse Childhood Experiences (ACEs) are potentially traumatic occurrences that happen during childhood from ages 0 to 17 [1,2]. The ACEs survey assesses childhood exposures to unique types of neglect, abuse, and household problems involving substance abuse, mental health, divorce, incarceration, suicide, and death [1]. Unhealthy coping behaviors are mechanisms of ACEs that lead to the onset of disease [3].

High ACEs are significantly linked to several harmful health behaviors consisting of physical inactivity [4], alcohol abuse [5], smoking [6], drug abuse [7], and attempted suicide [8]. Given the harmful nature of these coping mechanisms, ACEs are associated with an increased risk of chronic disease and premature death [9]. Chronic diseases associated with high ACEs include obesity [10], diabetes [11], heart disease [12], and cancer [13]. Mental health problems related to high ACEs include low life satisfaction [14], personality disorders [15], anxiety [14], and depression [16].

Depression is a common but serious mood disorder causing severe symptoms (*e.g.*, feeling sad, hopeless, irritable, disinterested and/or tired) that affect feelings, thoughts, and activities of daily living [17]. Approximately 7.8 percent (19.4 million) of U.S. adults experienced depression at some point in 2019 [18]. In the U.S., depression rates are highest among females, individuals of two or more races, and adults aged 18–25 years old [18]. The number of ACEs a person encounters significantly predicts their risk for depression in adulthood [19–21]. Individuals experiencing depression are more prone to engage in behaviors that are harmful to their health [22]. Some unhealthy behaviors associated with depression include disordered eating behaviors [23,24], low fruit and vegetable intake [25], physical inactivity [26], smartphone addiction [27], smoking [28], and the abuse of alcohol and other drugs [29,30]. Individuals living with depression are at an increased risk of comorbidities and chronic diseases [31]. Depression is a risk factor for poor sleep quality [32], obesity [33], heart disease [34], cancer [35], and suicide [36]. It is also noteworthy that a dose-response relationship exists between ACEs score and depression, as the risk for depression increases with each additional ACE [19,20,37].

There is an urgent need to identify protective factors that can effectively mitigate the impact of ACEs on depression later in life. A known protective factor against the effect of ACEs on depression is perceived social support, as adults with high ACEs and moderate-to-strong perceived social support have significantly lower odds of depression compared those with low perceived social support [38]. Emotional support has also been shown to protect against the deleterious effects of ACEs on depression [39]. Additionally, adults with ≥4 ACEs that grew up with an adult who fostered a sense of protection and safety have a lower likelihood of reporting mental health problems [40]. Depression research among individuals with high ACEs could benefit from including measures of physical activity, given the role physical activity can play in alleviating depression among adults [41]. To achieve optimal health benefits, adults should partake in at least 150-to-300 minutes of moderate-intensity (*e.g.*, brisk walking, yard work) physical activity each week [42]. Engaging in routine physical activity is an effective approach to prevent poor health outcomes including obesity [43], hypertension [44], diabetes [45], heart disease [46], and cancer [47]. Regular physical activity benefits brain and cognitive

health [48], promotes high-quality sleep [49], minimizes negative stress [50], and protects against anxiety and depression [41,51].

The original ACEs study highlighted physical inactivity as a risk factor for morbidity and mortality, but did not examine the relationship between ACEs and physical activity [4]. Uncertainty remains concerning the link between ACEs and physical activity, as research findings from an array of studies offer contradicting evidence [52]. For example, results from a multi-country study among young adults in eastern and central Europe reported greater odds of physical inactivity among young-adults with ≥4 ACEs compared to those with no ACEs [8]. Outcomes from a study among U.S. adults also yielded findings showing lower physical activity among individuals with ACEs compared to those with no ACEs [53]. Contrarily, research findings from studies conducted among adults in the U.S. state of Hawaii [54], England [55], Saudi Arabia [56], and the U.K. [13,57] suggest a non-significant relationship between ACEs and physical activity. Since physical activity has been shown to alleviate depression [41], there is a need to determine the extent to which physical activity mitigates the relationship between ACEs and depression among U.S. adults, as few studies have focused on this unique issue. A cross-sectional study among European adults ages 50-and-older produced results showing physical activity attenuated the link between ACEs and depression [58]. A longitudinal study in the U.S. yielded findings suggesting participation in team sports during adolescence attenuated the effect of ACEs on depressive symptoms later in life [59].

### Study aims and hypotheses

No studies in the U.S. have examined the protective nature of physical activity as it relates to the link between ACEs and depression. Therefore, the primary aims of this study were to address a knowledge gap concerning the link between ACEs and physical activity among U.S. adults, and whether physical activity protects against ACEs increasing the odds of depression. The research hypotheses for this study included: (1) an inverse relationship will exist between ACEs score and physical activity, and (2) physical activity will attenuate a positive association between ACEs and depression. Study findings derived from this research could be integral in informing future interventions to prevent depression among adults with high ACEs.

## Methods

### Participant sample

Data from the 2020 Behavioral Risk Factor Surveillance System (BRFSS) [60] were used for a study examining cross-sectional relationships among the primary variables of ACEs, physical activity, and depression. The 2020 BRFSS represents the most recently available data of an annual cross-sectional study of U.S. adults (n = 401,958) that is conducted by the CDC. Each year, the CDC partners with health agencies from all U.S. states and territories to conduct the BRFSS by surveying a representative sample of adults and assessing a variety of social conditions, health behaviors, and the disease histories.

Fewer than half of U.S. states and no U.S. territories collected ACEs data for the 2020 BRFSS; as such, the sample for this study included U.S. adults providing complete data for variables of interest (n = 117,204) from the District of Columbia (DC) and the following 21 states: Alabama, Arizona, Florida, Georgia, Hawaii, Idaho, Iowa, Kentucky, Mississippi, Missouri, Montana, Nevada, North Dakota, Rhode Island, South Carolina, South Dakota, Texas, Utah, Virginia, Wisconsin, and Wyoming. Adults of all ages (18+ years) were included in our sample to best evaluate the relationship between ACEs, physical activity, and depression across the U.S. adult lifespan.

## Measures

In the 2020 BRFSS, ACEs were measured using an 11-item ACEs survey capturing self-reported childhood exposures to unique types of neglect, abuse, and family problems involving substance abuse, mental health, divorce, incarceration, suicide, and death. Total ACEs score represented the primary predictor variable in this study. The 2020 BRFSS data for ACEs were separated by each of the 11 individual items. Individual ACEs items cover various types of childhood adversity; some of which include substance abuse in the home ("Did you live with anyone who used illegal drugs or who abused prescription medications?"), witnessing domestic violence ("How often did adults in your home ever slap, hit, kick, punch or beat each other up?"), physical abuse ("How often did an adult physically hurt you in any way?"), psychological abuse ("How often did an adult in your home ever swear at you, insult you, or put you down?"), and sexual abuse ("How often did anyone at least 5 years older than you touch you sexually?"). Researchers created an interval variable for the full ACEs score (0–11) by summing the total affirmative (once, more than once) responses for each of the 11 items.

Physical activity was measured in BRFSS 2020 with one item asking respondents if they engaged in physical activity or exercise during the past 30 days outside of any activity related to their job (no = 0, yes = 1). For this study, physical activity was separately modeled as a predictor of depression and an outcome of ACEs. Depression was also measured in BRFSS 2020 with one item in by asking respondents if they had ever been diagnosed with a depressive disorder including dysthymia, minor depression, depression, or major depression (no = 0, yes = 1). For this study, depression was treated as the primary outcome variable. Both physical activity and depression were dichotomous variables.

Covariates included variables from the 2020 BRFSS data for age group (18–24, 25–34, 35–44, 45–54, 55–64, and 65+ years), sex (female, male), race/ethnicity (American Indian/Alaska Native, Asian, Black, Hispanic, Native Hawaiian/Pacific Islander, White, multiple races, and other), and income (<$15k, $15–24.9k, 25–34.9k, 35–49.9k, ≥$50k, and unsure). Information on the validity of the primary study variable measures of ACEs, physical activity, and depression has been previously reported [61]. Listwise deletion was used to remove cases with missing data on the primary variables of ACEs, physical activity, and depression, which reduced the sample size from n = 401,958 to n = 117,204.

## Statistical analysis

RStudio [62] packages including 'stats', 'glm', and 'lmer' were used to analyze the study data. Due to the sample clustering by state, intraclass correlations (ICC) were calculated using unconditional random intercept models to test the extent to which clustering altered the study outcomes of interest. An ICC of >0.05 would have justified the use of a mixed-effects multilevel model [63], but fixed-effects general linear models (GLM) using multivariable logistic regression were sufficient for estimating accurate effect size coefficients for our primary variables of interest [64]. An additional analysis was run using a dichotomous variable for missingness to test if participants excluded from study due to missing data (n = 284,754) were systematically different in their characteristics from participants included in the study with complete data (n = 117,204).

Pearson's chi-square tests were conducted to analyze the relationship between having ≥4 ACEs and either physical activity, depression, or the covariates. Regression models were fitted for predictors (ACEs or physical activity) and outcomes (physical activity or depression) while adjusting for all covariates. An ACEs-Physical Activity interaction term was tested to determine whether physical activity attenuated the relationship between ACEs and depression [65]. Adjusted odds ratios (AOR) were computed by exponentiating the unstandardized beta

coefficients for the effect of a predictor on an outcome [66]. Statistical analyses estimated the links between ACEs and depression, ACEs and physical activity, and physical activity and depression. An interaction effect of ACEs-and-physical activity on depression was tested to determine if physical activity protects against ACEs increasing the odds of depression. AOR's were produced for predictors in each model. Participant characteristics were modeled as covariates in all analyses.

## Results

The study sample consisted of adults (n = 117,204) ages 18-and-older from 21 U.S. states and DC (Table 1). Characteristics for participants included in the study differed in sex, age group, race/ethnicity, and income ($p<0.0001$) compared to excluded participants. Clustering by state did not alter study outcomes for physical activity (ICC = 0.01) nor depression (ICC = 0.005). Mean participant age was 55.3 years (SD 17.7). Most adults were 65+ years old (37.3%), female (54.8%), White (75.2%), and reported an income of $\geq$$50,000.

A majority of adults had a total of <4 ACEs (83%), engaged in physical activity (75.7%), and had never been diagnosed with depression (81.5%). Results from the chi-square analyses (Table 1) indicated that having $\geq$4 ACEs is related to physical activity ($\chi^2$ = 19.4, df = 1; $p<0.01$), depression ($\chi^2$ = 6,841.6, df = 1; $p<0.0001$), sex ($\chi^2$ = 312.7, df = 1; $p<0.0001$), age ($\chi^2$ = 4,485.9, df = 5; $p<0.0001$), race/ethnicity ($\chi^2$ = 997.8, df = 7; $p<0.0001$), and income ($\chi^2$ = 1,359.1, df = 5; $p<0.0001$).

Across all sampled U.S. adults, findings from the multivariable regression models (Table 2) showed ACEs increased the odds of depression (AOR = 1.050; 95% CI = 1.049, 1.051), as the percentage of depressed U.S. adults increased by 5% for each additional ACE ($p<0.001$).

ACEs were inversely associated with physical activity (AOR = 0.994; 95% CI = 0.992, 0.995); with the percentage of U.S. adults engaging in physical activity dropping 1% for each additional ACE ($p = <0.001$). Physical activity was inversely associated with depression (AOR = 0.927; 95% CI = 0.922, 0.932), which highlights the odds for depression were roughly 7% lower for U.S. adults reporting physical activity ($p<0.001$).

Results from an additional analysis with the ACEs-Physical Activity interaction term (Fig 1) indicate the odds of ACEs being linked to depression are lower among adults reporting physical activity in the past 30 days (AOR = 0.995; 95% CI = 0.993, 0.996) compared to those not reporting physical activity, as physical activity reduced the odds of ACEs contributing to depression ($p<0.001$).

## Discussion

Nearly one-fifth of the U.S. adults included in this study sample had $\geq$4 ACEs. Disparities in unhealthy behavior and chronic disease persist among individuals with $\geq$4 ACEs compared to those with <4 ACEs [52]. Individuals with $\geq$4 ACEs are at a much greater risk of premature mortality due to the increased risk of developing various diseases [67]. The numerous U.S. adults with $\geq$4 ACEs are at a disadvantage concerning their life health trajectories.

Findings from this study support existing research evidence concerning ACEs increasing the likelihood of depression [16,19–21,37]. Previous research exploring the mechanisms of ACEs have determined resilience [68,69], self-esteem [70,71], mindfulness [72], stress [69,73,74], inflammation [75], intimate partner violence [73], and physical activity [58] play significant roles in the relationship between ACEs and depression. Results of this study along-side previous findings could be useful in helping inform intervention designs to prevent depression among adults with high ACEs.

**Table 1. Participant characteristics and Descriptives by Adverse Childhood Experiences (ACEs) among U.S. Adults (n = 117,204).**

| Characteristics (M, SD)[a] | Total (%) | <4 ACEs (%) | ≥4 ACEs (%) | $\chi^2$ |
|---|---|---|---|---|
| Sample size (%) | 117,204 (100) | 97,314 (83) | 19,890 (17) | |
| *Sex* | | | | 312.7* |
| Female | 64,238 (54.8) | 52,205 (53.6) | 12,033 (60.5) | |
| Male | 52,966 (45.2) | 45,109 (46.4) | 7,857 (39.5) | |
| Age in years (M = 55.3, SD = 17.7) | | | | 4,485.9* |
| 18–24 | 7,190 (6.1) | 5,321 (5.5) | 1,869 (9.4) | |
| 25–34 | 11,977 (10.2) | 8,625 (8.9) | 3,352 (16.9) | |
| 35–44 | 14,957 (12.8) | 11,323 (11.6) | 3,634 (18.3) | |
| 45–54 | 17,050 (14.5) | 13,450 (13.8) | 3,600 (18.1) | |
| 55–64 | 22,338 (19.1) | 18,587 (19.1) | 3,751 (18.8) | |
| 65+ | 43,692 (37.3) | 40,008 (41.1) | 3,684 (18.5) | |
| *Race/Ethnicity* | | | | 997.8* |
| American Indian/Alaska Native | 2,311 (2) | 1,638 (1.7) | 673 (3.4) | |
| Asian | 3,177 (2.7) | 2,949 (3) | 228 (1.2) | |
| Black | 10,158 (8.7) | 8,435 (8.7) | 1,723 (8.7) | |
| Hispanic | 8,991 (7.7) | 7,095 (7.3) | 1,896 (9.5) | |
| Native Hawaiian/Pacific Islander | 700 (0.6) | 532 (0.5) | 168 (0.8) | |
| White | 88,203 (75.2) | 74,027 (76.1) | 14,176 (71.3) | |
| Multiple Races | 2,844 (2.4) | 1,988 (2) | 856 (4.3) | |
| Other | 820 (0.7) | 650 (0.7) | 170 (0.8) | |
| *Income* | | | | 1,359.1* |
| <$15,000 | 7,838 (6.7) | 5,723 (5.9) | 2,115 (10.6) | |
| $15,000–$24,999 | 15,020 (12.8) | 11,691 (12) | 3,329 (16.7) | |
| $25,000–$34,999 | 9,862 (8.4) | 7,946 (8.2) | 1,916 (9.6) | |
| $35,000–$49,999 | 13,926 (11.9) | 11,472 (11.8) | 2,454 (12.3) | |
| ≥$50,000 | 52,291 (44.6) | 44,376 (45.6) | 7,915 (39.9) | |
| Don't Know | 18,267 (15.6) | 16,106 (16.5) | 2,161 (10.9) | |
| *Physical Activity in past 30 days* | | | | 19.4* |
| No | 28,535 (24.3) | 23,449 (24.1) | 5,086 (25.6) | |
| Yes | 88,669 (75.7) | 73,865 (75.9) | 14,804 (74.4) | |
| *Depression* | | | | 6,841.6* |
| No | 95,540 (81.5) | 83,453 | | |
| Yes | 21,664 (19.5) | 13,861 | | |

[a]Mean, Standard Deviation.

*$p<0.01$.

Outcomes in this study involving physical activity provide important evidence to address general uncertainty about the link between ACEs and physical activity. Contradictory results have been accumulated from past studies examining the extent to which ACEs are related to physical activity [52]. The two previous studies in the U.S. that assessed ACEs and physical activity produced discordant findings: one study used BRFSS data collected in 2009–2012 from 14 states and yielded findings suggesting a significant link between ACEs and physical activity [53], while the other study used BRFSS data collected in 2010 from the state of Hawaii and produced non-significant findings for the relationship between ACEs and physical activity [54]. BRFSS data for this study were collected in 2020 from 22 U.S. states, and results reported here offer an expansion of the knowledge base by emphasizing how each additional ACE significantly reduces the likelihood of U.S. adults engaging in physical activity.

**Table 2. Statistical models and results for the Associations between Adverse Childhood Experiences (ACEs), physical activity, and depression among U.S. adults (n = 117,204).**

|  | Model[a] | AOR[b] | 95% CI[c] | p-value |
|---|---|---|---|---|
| Model 1 | ACEs→Depression | 1.05 | 1.049, 1.051 | p<0.001 |
| Model 2 | ACEs→Physical Activity | 0.994 | 0.992, 0.995 | p<0.001 |
| Model 3 | Physical Activity→Depression | 0.927 | 0.922, 0.932 | p<0.001 |
| Model 4 | ACEs×Physical Activity→Depression | 0.995 | 0.993, 0.996 | p<0.001 |

[a]All multivariable logistic regression analyses adjusted for age, sex, race/ethnicity, and income.

[b]Adjusted Odds Ratio.

[c]95% confidence interval.

The findings in this study also filled a critical knowledge gap concerning the extent to which physical activity protects against ACEs increasing the probability of depression among U.S. adults. The only other study to analyze this relationship was conducted among adults ages 50-and-older from several European countries, and also determined that physical activity provided significant protection against the effect of ACEs on depression [58]. Past intervention studies have detected a dose-response relationship by using physical activity to treat depression, as depressive symptoms decreased as physical activity increased [76]. Results from meta-analyses detailing effects of physical activity on depression in both observational and

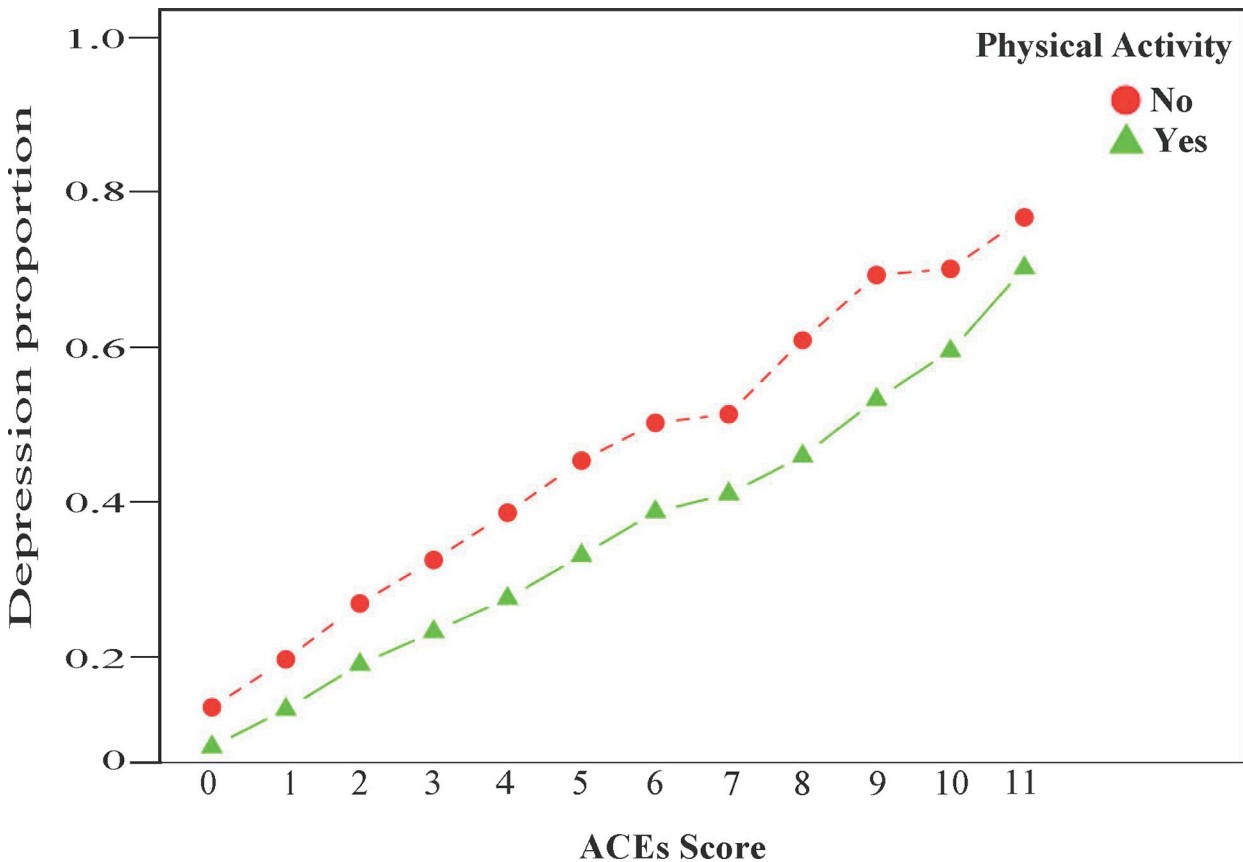

**Fig 1. Interaction plot demonstrating physical activity attenuating the relationship between ACEs and depression among U.S. adults (n = 117,204).** [a]Model displayed in figure adjusted for age, sex, race/ethnicity, and income.

intervention studies reported moderate-to-large negative effects of physical activity on depression symptoms [77,78], which underscores the possibility of physical activity being a promising approach to alleviate increased rates of depression among adults with high ACEs. Our study findings revealed that ACEs decreased the likelihood of physical activity, while also suggesting physical activity decreased the likelihood of depression. Moreover, these findings suggest physical activity reduced the probability of ACEs contributing to depression.

A strength of this study pertains to the large sample size, as it was possible to accurately estimate cross-sectional effects for the primary variables of interest among a representative sample of U.S. adults. Another strength is the use of the Kaiser-CDC measure for ACEs, which is considered the gold-standard measure of ACEs. Another key strength includes how these study findings help to address a knowledge gap concerning the link between ACEs, physical activity, and depression.

This study had several limitations. One limitation concerns the preexisting nature of the data and the researchers' lack of control over the instruments used to measure primary variables. In particular, measures for depression and physical activity were limited in their comprehensiveness. The depression measure used only one item that focused on a past diagnosis of depression and did not evaluate current depressive symptoms. The physical activity measure also used just one item that asked about engaging in physical activity in the past 30 days, which does not provide sufficient information for daily or weekly physical activity levels to determine whether participants are meeting the recommended level of regular physical activity. Another limitation of this study was the absence of a pharmacologic treatment variable to statistically adjust for any confounding effects of prescription drug use on depression outcomes. All measures were limited in their recall bias, as some respondents to the BRFSS 2020 survey may have had trouble accurately recollecting important information concerning ACEs occurrences, physical activity frequency, and depression diagnoses. An additional limitation includes the cross-sectional nature of this study, as researchers could not establish temporal relationships using multiple timepoints for each variable to determine the extent to which ACEs change the odds of physical activity and depression over time. A detail that could be perceived as a limitation involves the small, yet statistically significant odds ratios reported in the results. The dichotomous nature of the physical activity and depression variables contain floor and ceiling boundaries for the effect size that limits their maximum change in odds to ±1. It is necessary to highlight that the BRFSS 2020 data for physical activity and depression were collected as dichotomous outcomes which requires researcher to examine percentage changes in yes/no proportions (0–100%) and not larger coefficients from a broader continuous scale when conducting odds ratio analyses. Lastly, another limitation involved how systematic error was introduced in excluding participants with missing data on ACEs, physical activity, and/or depression, as regression analyses determined characteristics of included participants differed from excluded participants.

Despite these limitations, evidence from this study has the potential to inform future longitudinal research examining the extent to which physical activity partially mediates the effect of ACEs on depression among adults. Such research could eventually inform intervention studies testing the effectiveness of a physical activity treatment in preventing depression among U.S. adults with high ACEs. Physical activity interventions have been shown to successfully alleviate depression among adults [79]. The evidence from this study and past studies [16,19–21,37] demonstrate ACEs increase the odds of depression. It is necessary to apply a physical activity intervention among adults with high ACEs to examine the extent to which increased physical activity changes depression outcomes over time. Such research would provide important insight detailing potentially useful approaches for preventing depression among at-risk adults with high ACEs.

## Conclusions

ACEs pose serious risks to U.S. adults for increased odds of depression and reduced odds of physical activity. Findings from this study provide insight related to the potential for physical activity as an effective approach to lower the likelihood of depression among vulnerable adults with high ACEs. Future studies should longitudinally test physical activity as a mediator within the effect of ACEs on depression, which could ultimately lead to testing physical activity interventions as an approach to prevent depression among adults with high ACEs. Physical activity seems to be a promising approach for alleviating the heightened depression levels of adults with high ACEs.

## Acknowledgments

We thank Dr. David MacKinnon for providing guidance on the research methodology and statistical analyses, while also providing useful feedback on the manuscript.

## Author Contributions

**Conceptualization:** Michael F. Royer.

**Data curation:** Michael F. Royer.

**Formal analysis:** Michael F. Royer.

**Investigation:** Michael F. Royer.

**Methodology:** Michael F. Royer.

**Project administration:** Michael F. Royer.

**Resources:** Michael F. Royer.

**Software:** Michael F. Royer.

**Supervision:** Michael F. Royer, Christopher Wharton.

**Visualization:** Michael F. Royer.

**Writing – original draft:** Michael F. Royer.

**Writing – review & editing:** Michael F. Royer, Christopher Wharton.

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
