## [Decision Letter · Decision Letter 0]

24 Aug 2022

PONE-D-22-17743Physical activity mitigates the link between Adverse Childhood Experiences and Depression among U.S. adultsPLOS ONE

Dear Dr. Royer,

Thank you for submitting your manuscript to PLOS ONE. After careful consideration, we feel that it has merit but does not fully meet PLOS ONE’s publication criteria as it currently stands. Therefore, we invite you to submit a revised version of the manuscript that addresses the points raised during the review process.

 Please submit your revised manuscript by Oct 08 2022 11:59PM. If you will need more time than this to complete your revisions, please reply to this message or contact the journal office at plosone@plos.org. Please include the following items when submitting your revised manuscript:A rebuttal letter that responds to each point raised by the academic editor and reviewer(s). You should upload this letter as a separate file labeled 'Response to Reviewers'.A marked-up copy of your manuscript that highlights changes made to the original version. You should upload this as a separate file labeled 'Revised Manuscript with Track Changes'.An unmarked version of your revised paper without tracked changes. You should upload this as a separate file labeled 'Manuscript'.If applicable, we recommend that you deposit your laboratory protocols in protocols.io to enhance the reproducibility of your results. Protocols.io assigns your protocol its own identifier (DOI) so that it can be cited independently in the future. For instructions see: https://journals.plos.org/plosone/s/submission-guidelines#loc-laboratory-protocols. Additionally, PLOS ONE offers an option for publishing peer-reviewed Lab Protocol articles, which describe protocols hosted on protocols.io. Read more information on sharing protocols at https://plos.org/protocols?utm_medium=editorial-email&utm_source=authorletters&utm_campaign=protocols.

We look forward to receiving your revised manuscript.

Kind regards,

Catalina Castaño, M.D, MSc

Guest Editor

PLOS ONE

Journal Requirements:

2. Please ensure that you include a title page within your main document. You should list all authors and all affiliations as per our author instructions and clearly indicate the corresponding author.

Additional Editor Comments: 

Your article is significant because it addresses the importance of non-medical interventions in preventing mental health pathology. Your paper is well written, but I agree with Reviewer Nº1 that the abstract should be rewritten under his suggestions, focusing on the results. 

Overall I agree with the reviewer's suggestion; I think that background, methods and results are sound and wouldn´t make significant changes. Still, I have the same question as reviewer Nº2: Was pharmacologic treatment (previous or ongoing) a survey variable? Did they undergo or were under therapy? I think you should address that issue under discussion or limitations.

Reviewers' comments:

Reviewer's Responses to Questions

**Comments to the Author**

1. Is the manuscript technically sound, and do the data support the conclusions?

Reviewer #1: Yes

Reviewer #2: Yes

2. Has the statistical analysis been performed appropriately and rigorously? 

Reviewer #1: Yes

Reviewer #2: Yes

3. Have the authors made all data underlying the findings in their manuscript fully available?

Reviewer #1: Yes

Reviewer #2: Yes

4. Is the manuscript presented in an intelligible fashion and written in standard English?

Reviewer #1: Yes

Reviewer #2: Yes

5. Review Comments to the Author

Reviewer #1: Dear Authors,

Thank you for your excellent effort to publish your research work, it is an important study. Here are some comments and suggestions for you.

Starting from keywords, you are listed many words and decrease to four to six words, and use only relevant words.

In the abstract, you did not write the main objective and significance of the study which is very important to write in the background under the abstract. In addition, you took the content of the abstract directly from the body part of your research. But, it should be paraphrased or other similar words.

Introduction is good. But, in the methodology, the sampling techniques and procedures did not clearly state. so, make it clear. The result, discussion, and conclusion are good. But, state them in a short and precise way.

Reviewer #2: The article is relevant and highlights observable facts in all disciplines of medicine that have been minimized or "denied" by clinics that relegate such observations. It suggests that due to possible difficulties of culture and researchers, which historically tends to normalize damage to the health of minors in neglectful or violent and vulnerable environments. In this order of ideas, the contribution of this article gives us excellent support on how epidemiology is a good way of working for health and prevention. It was a pleasure to read and observe this collaborative work of the epidemiological surveillance centers of so many states that corroborates what is in our daily clinical practice with a methodological and statistical clarity of great help for future developments and treatments; with an n that gives an excellent value to the research for its diversity of populations in a single study. Adverse childhood experiences, depression, and exercise show new directions in holistic medicine and collaborative research. Each one is a subject as a variable with higher risk factors for many diseases and higher mortality. If we study it together and understand how each works to increase or decrease healthy living, we can help people in a better way. The limitations are well established and explained, with the suggestion of prospective longitudinal studies in the future. Literature is appropriate and fair. I suggest adding new chapter studies with a representative in vivo sample (as a gold standard) will reinforce the results. It does recommend adding an evaluation of the subjects by the clinician to know the reliable or specific depression diagnostic instruments.

Question: How many people are taking medicine during the survey? Or the exercise? It is a posible confusional variable? Are the authors looking for it?

Evidence recommend being careful with this vital variable if working around depressive people and looking to probe any therapy. It is an ethical and methodologic mandatory issue. Please clarify.

6. PLOS authors have the option to publish the peer review history of their article (what does this mean?). If published, this will include your full peer review and any attached files.

Reviewer #1: **Yes: **Beshir Mammiyo

Reviewer #2: **Yes: **ADELINA ALCORTA-GARZA

---

## [Author Response · Author response to Decision Letter 0]

25 Aug 2022

Editor Comments:

1. Was pharmacologic treatment (previous or ongoing) a survey variable? Did they undergo or were under therapy? I think you should address that issue under discussion or limitations.

Response 1:

Thank you for emphasizing this consideration. The only data for pharmacologic treatment included within the BRFSS dataset was for individuals with a history of cancer or hepatitis B. Please see lines 233-235 of the Discussion section for mention of the lack of a pharmacologic treatment variable as a study limitation.

Reviewer 1 Comments:

1. Starting from keywords, you are listed many words and decrease to four to six words, and use only relevant words.

Response 1:

Thank you for this feedback. I have now decreased the number of keywords to five key words.

2. In the abstract, you did not write the main objective and significance of the study which is very important to write in the background under the abstract. 

Response 2:

Thank you for this suggestion. Please see the modifications made to the Abstract, which now includes mention of the study objective and significance. 

3. In addition, you took the content of the abstract directly from the body part of your research. But, it should be paraphrased or other similar words.

Response 3:

Edits have now been made to the Abstract in an effort to make the abstract unique from similar content in the main manuscript text.

4. Introduction is good. But, in the methodology, the sampling techniques and procedures did not clearly state. so, make it clear. The result, discussion, and conclusion are good. But, state them in a short and precise way.

Response 4:

Please refer to lines 73-87 concerning the sampling techniques and procedures. Since the BRFSS is an archival dataset, sampling approaches were restricted to including/excluding participant data from the full BRFSS dataset according to whether the state health agency collected data for ACEs, physical activity, and/or depression. Lines 115-117 detail the listwise deletion approach used to remove missing data, which determined the final sample size for our study.

Reviewer 2 Comments:

1. I suggest adding new chapter studies with a representative in vivo sample (as a gold standard) will reinforce the results. 

Response 1:

Thank you for this suggestion. Lines 208-216 highlight the only known studies (some in vivo, some not) that have previously examined the links between ACEs and depression, ACEs and physical activity, and physical activity and depression. Lines 208-210 specifically discuss the only known study assessing the role of physical activity within the link between ACEs and depression.

2. It does recommend adding an evaluation of the subjects by the clinician to know the reliable or specific depression diagnostic instruments.

Response 2:

This is certainly an important consideration. Unfortunately, the BRFSS resource page on CDC.gov does not contain information for the specific depression diagnostic instruments used to inform the depression-related items used for the 2020 BRFSS. Please see lines 226-233 in the discussion detailing the limitations of the one-item instrument used to measure both depression and physical activity.

3. Question: How many people are taking medicine during the survey? Or the exercise? It is a posible confusional variable? Are the authors looking for it?

Response 3:

Thank you for bringing attention to this important detail. The only data for participants taking medicine during the survey that is included within the BRFSS dataset is for individuals with a history of cancer or hepatitis B. Please see lines 233-235 of the Discussion section for mention of the lack of a pharmacologic treatment variable as a study limitation.

---

## [Decision Letter · Decision Letter 1]

12 Sep 2022

Physical activity mitigates the link between adverse childhood experiences and depression among U.S. adults

PONE-D-22-17743R1

Dear Dr. Royer,

We’re pleased to inform you that your manuscript has been judged scientifically suitable for publication and will be formally accepted for publication once it meets all outstanding technical requirements.

Kind regards,

Catalina Castaño, M.D, MSc

Guest Editor

PLOS ONE

Additional Editor Comments (optional):

Reviewers' comments:

Reviewer's Responses to Questions

**Comments to the Author**

1. If the authors have adequately addressed your comments raised in a previous round of review and you feel that this manuscript is now acceptable for publication, you may indicate that here to bypass the “Comments to the Author” section, enter your conflict of interest statement in the “Confidential to Editor” section, and submit your "Accept" recommendation.

Reviewer #1: All comments have been addressed

2. Is the manuscript technically sound, and do the data support the conclusions?

Reviewer #1: Yes

3. Has the statistical analysis been performed appropriately and rigorously? 

Reviewer #1: Yes

4. Have the authors made all data underlying the findings in their manuscript fully available?

Reviewer #1: Yes

5. Is the manuscript presented in an intelligible fashion and written in standard English?

Reviewer #1: Yes

6. Review Comments to the Author

Reviewer #1: It was already commented and all comments were addressed. It is presented with a good fashion and the tittle is sound able

7. PLOS authors have the option to publish the peer review history of their article (what does this mean?). If published, this will include your full peer review and any attached files.

Reviewer #1: **Yes: **Mammiyo Beshir

---

## [Editor Report · Acceptance letter]

14 Sep 2022

PONE-D-22-17743R1 

Physical activity mitigates the link between adverse childhood experiences and depression among U.S. adults. 

Dear Dr. Royer:

I'm pleased to inform you that your manuscript has been deemed suitable for publication in PLOS ONE. Congratulations! Your manuscript is now with our production department. 

Kind regards, 

on behalf of

Dr. Catalina Castaño 

Guest Editor

PLOS ONE